# Chemical Constituents and Anti-Angiogenic Principles from a Marine Algicolous *Penicillium sumatraense* SC29

**DOI:** 10.3390/molecules27248940

**Published:** 2022-12-15

**Authors:** Hsiao-Yang Hsi, Shih-Wei Wang, Chia-Hsiung Cheng, Ka-Lai Pang, Jyh-Yih Leu, Szu-Hsing Chang, Yen-Tung Lee, Yueh-Hsiung Kuo, Chia-Ying Huang, Tzong-Huei Lee

**Affiliations:** 1Institute of Fisheries Science, National Taiwan University, Taipei 10617, Taiwan; 2Institute of Biomedical Sciences, MacKay Medical College, New Taipei City 252, Taiwan; 3Department of Medicine, MacKay Medical College, New Taipei City 252, Taiwan; 4Graduate Institute of Natural Products, College of Pharmacy, Kaohsiung Medical University, Kaohsiung 807, Taiwan; 5Department of Biochemistry and Molecular Cell Biology, School of Medicine, College of Medicine, Taipei Medical University, Taipei 11031, Taiwan; 6Graduate Institute of Medical Science, College of Medicine, Taipei Medical University, Taipei 11031, Taiwan; 7Graduate Institute of Cancer Biology and Drug Discovery, College of Medical Science and Technology, Taipei Medical University, Taipei 11031, Taiwan; 8Institute of Marine Biology and Centre of Excellence for the Oceans, National Taiwan Ocean University, Keelung 202301, Taiwan; 9The Department of Life Science, Fu Jen Catholic University, New Taipei City 242304, Taiwan; 10Graduate institute of Applied Science and Engineering, College of Science and Engineering, Fu Jen Catholic University, New Taipei City 242304, Taiwan; 11Department of Chinese Medicine, MacKay Memorial Hospital, Taipei 10449, Taiwan; 12Department of Cosmetic Science, College of Human Ecology, Chang Gung University of Science and Technology, Taoyuan 33303, Taiwan; 13Department of Chinese Pharmaceutical Sciences and Chinese Medicine Resources, China Medical University, Taichung 40447, Taiwan; 14Department of Biotechnology, Asia University, Taichung 41354, Taiwan; 15Chinese Medical Research Center, China Medical University, Taichung 40447, Taiwan

**Keywords:** *Penicillium sumatraense*, penisterine, anti-angiogenesis, endothelial progenitor cells, zebrafish

## Abstract

In this study, a marine brown alga *Sargassum cristaefolium*-derived fungal strain, *Penicillium sumatraense* SC29, was isolated and identified. Column chromatography of the extracts from liquid fermented products of the fungal strain was carried out and led to the isolation of six compounds. Their structures were elucidated by spectroscopic analysis and supported by single-crystal X-ray diffraction as four previously undescribed (*R*)-3-hydroxybutyric acid and glycolic acid derivatives, namely penisterines A (**1**) and C–E (**3**–**5**) and penisterine A methyl ether (**2**), isolated for the first time from natural resources, along with (*R*)-3-hydroxybutyric acid (**6**). Of these compounds identified, penisterine E (**5**) was a unique 6/6/6-tricyclic ether with an acetal and two hemiketal functionalities. All the isolates were subjected to in vitro anti-angiogenic assays using a human endothelial progenitor cell (EPCs) platform. Among these, penisterine D (**4**) inhibited EPC growth, migration, and tube formation without any cytotoxic effect. Further, in in vivo bioassays, the percentages of angiogenesis of compound **3** on *Tg* (*fli1*:EGFP) transgenic zebrafish were 54% and 37% as the treated concentration increased from 10.2 to 20.4 µg/mL, respectively, and the percentages of angiogenesis of compound **4** were 52% and 41% as the treated concentration increased from 8.6 to 17.2 µg/mL, respectively. The anti-angiogenic activity of penisterine D (**4**) makes it an attractive candidate for further preclinical investigation.

## 1. Introduction

*Sargassum pallidum* and *S. fusiforme*, two species of marine brown algae, have long been used as traditional Chinese medicines for the treatments of phlegm elimination and detumescence [1]. With abundant and highly diversified bioactive secondary metabolites as allelochemicals or defensive strategy [2,3], the *Sargassum* spp. usually dominate in the subtidal zone of the Central Indo-Pacific region during the spring season [4]. It was also reported that the alga-associated microorganisms could exert remarkable effects to improve the algal ability to survive in harsh environments [5]. That implied the microorganisms derived from *Sargassum* spp. could be a promising source for bioactive natural products. So far, a number of secondary metabolites with anti-inflammatory activity have been disclosed from subtropical *Sargassum* spp., whereas studies on the tropical species focused mainly on the polysaccharides [3]. However, chemical investigations of both tropical and subtropical algicolous microorganisms have not been conducted intensively [5].

Cell growth and tube formation by endothelial progenitor cells (EPCs) are drivers of angiogenesis, which enable new blood vessels to develop in existing vasculature [6], and EPCs are players in cancer progression, and they contribute to both tumor germination and maintaining an inflammatory state [7]. Therefore, anti-angiogenic therapy has been employed as an anti-cancer strategy in recent years, aiming to block the growth of tumor blood vessels, thereby inhibiting tumor growth [8]. Moreover, the zebrafish model has been gradually applied to the studies of many diseases because of its easy observation characteristics [9]. Zebrafish have also been widely used in the tests of drug efficacy and toxicity and active substance screening. Friend leukemia integration 1 (*fli1*) is a gene closely related to angiogenesis, which is expressed in vascular endothelial cells. The *fli1:EGFP* recombinant gene sequence is inserted into the genome of *Tg (fli1:EGFP)* transgenic zebrafish, and under the regulation of the *fli1* promoter, the green fluorescent protein *EGFP* in this gene is expressed, and the endothelial cells of all blood vessels were fluorescent under the fluorescence microscope, which could be directly applied to observe the angiogenesis of zebrafish [10]. The *Tg (fli1:EGFP)* transgenic zebrafish has become an anti-angiogenic, high-throughput drug screening model and has been used to evaluate the anti-angiogenic activity of natural products [11].

Due to the location at tropical and subtropical regions, the resources of marine algae are abundant in Taiwan, and at least eight *Sargassum* species have been identified locally [12]. As they have been developed potentially for functional foods or dietary supplements during the past decade, fucose-containing sulfated polysaccharides (fucoidans) were considered to be the active principles of local *Sargassum* spp. and to exert a wide array of bioactivities, such as antioxidant, anti-inflammatory, antilipogenic, immune promoting, and anti-infection activities [13,14,15]. As in other tropical and subtropical regions, the chemical investigations of the local *Sargassum*-derived microorganisms still remain rare. Thus, efficient agar-based isolation, small-scale liquid fermentation, and screening by anti-angiogenic platform were performed sequentially for pursuing bioactive fungal strains from *S. cristaefolium* collected in Taiwan. In an attempt to unravel the bioactive principles of *P. sumatraense* SC29 isolated from *S. cristaefolium*, a series of fungal cultivation, compound separation, and structural determination was thus undertaken and resulted in the identification of five (*R*)-3-hydroxybutyric acid and glycolic acid derivatives **1**–**5** (Figure 1), together with (*R*)-3-hydroxybutyric acid. The in vitro and in vivo anti-angiogenic evaluation of **1**–**5** in human endothelial progenitor cells (EPCs) and embryonic zebrafish model were also performed.

## 2. Results and Discussion

### 2.1. Isolation and Characterization of Secondary Metabolites

In this study, the brown alga *S. cristaefolium*-derived fungal strain *P. sumatraense* SC29 was cultured in potato dextrose broth and malt extract, and six compounds—including four unreported (**1** and **3**–**5)** and the previously described penisterine A methyl ether (**2**), which was isolated for the first time from natural resource, along with (*R*)-3-hydroxybutyric acid (**6**)—were purified from the fermented products. Of these, (*R*)-3-hydroxybutyric acid has been reported to exhibit antibiotic activity and could possibly serve as a chiral building block for the synthesis of fine chemicals such as antibiotics, vitamins, aromatics, and pheromones [16]. It was also used as monomer to produce poly[(*R*)-3-hydroxybutyrate], a kind of biodegradable plastic with properties comparable to those of polypropylene [17].

Compound **1** was obtained as a colorless oil. The quasi-molecular ion peak [M − H]^−^ at *m/z* 161.0450 (calcd. 161.0450 for C_6_H_9_O_5_) in the HRESIMS and supported by ^13^C NMR of **1** (Table 1), indicating a molecular formula of C_6_H_10_O_5_. The IR spectrum indicated the presence of a carboxylic acid (3400–2400 cm^−1^) and a ketone carbonyl functionality (1728 cm^−1^). The ^1^H NMR spectrum of **1** (Table 2) showed signals of a methyl proton at δ_H_ 1.31 (3H, d, *J* = 6.6 Hz, H-6), two methylene protons at δ_H_ 2.56 (1H, dd, *J* = 16.2, 5.4 Hz, H_a_-4) and 2.63 (1H, dd, *J* = 16.2, 8.4 Hz, H_b_-4) and δ_H_ 4.07 (2H, br s, H-1), and a methine proton at δ_H_ 5.31 (1H, dqd, *J* = 8.4, 6.6, 5.4 Hz, H-3). The ^13^C NMR spectrum, coupled with DEPT and phase-sensitive HSQC, of **1** (Table 1) showed six carbon signals, including a methyl at δ_C_ 20.1 (C-6), two methylenes at δ_C_ 41.4 (C-4) and 61.2 (C-1), an oxygenated methine at δ_C_ 69.6 (C-3), and two carbonyl carbon signals at δ_C_ 173.8 (C-2) and 174.1 (C-5). Key cross-peaks from H_3_-6/H-3 and H-3/H_2_-4 in the COSY spectrum in combination with key cross-peaks from H_2_-4/C-5, H-3/C-2, and H_2_-1/C-2 in the HMBC spectrum (Figure 2) established the gross structure of **1** as shown. Alkaline hydrolysis of **1**, followed by HPLC purification, gave 3-hydroxybutyric acid (**6**) as evidenced by comparing its ^1^H NMR and ^13^C NMR (Appendix A) with those of (*R*)-3-hydroxybutyric acid (**6**). The sign of optical rotation ([α]^26^_D_ = −20.2) of **6** was consistent with that of (*R*)-3-hydroxybutyric acid ([α]^25^_D_ = −16.0) in the literature [18]. The configuration of C-3 in **1** was thus determined to be *R*.

The spectroscopic data of **2** resembled those of **1** except for the presence of an additional methyl group (δ_H_ 3.67s/ δ_C_ 52.4) (Table 1 and Table 2). The additional methyl group was assigned to be attached at OH-5 of **1** to form a methoxyl moiety on **2** based on a key cross-peak of H_3_-7/C-5 in the HMBC spectrum of **2** (Figure 2). Thus, compound **2** was determined to be the methyl analogue of **1**. In the Scifinder^n^ database (Chemical Abstracts Service, American Chemical Society, Columbus, Ohio, USA), it was shown that **2** seemed to be a purchasable chemical (No. 1841321-02-5); however, no reference and spectroscopic data were provided. Therefore, we present the ^1^H and ^13^C NMR data of **2**.

Compound **3**, obtained as a brown oil, was determined to have a molecular formula of C_9_H_16_O_5_, as evidenced by its HRESIMS analysis and ^13^C NMR spectrum (Table 1). The IR absorption band at 1739 cm^−1^ indicated the presence of a ketone carbonyl group. The ^1^H NMR data along with the HSQC spectrum of **3** showed a methyl signal at δ_H_ 1.31 (d, *J* = 6.6 Hz, H_3_-6), three methoxyl signals at δ_H_ 3.18 (s, H_3_-7), 3.31 (s, H_3_-8), and 3.39 (s, H_3_-9), a set of nonequivalent methylene signals at δ_H_ 2.36 (dd, *J* = 13.5, 3.0 Hz, H_a_-2) and 2.65 (dd, *J* = 13.5, 11.4 Hz, H_b_-2), and two oxygenated methine signals at δ_H_ 4.11 (dqd, *J* = 11.4, 6.6, 3.0 Hz, H-1) and 4.94 (s, H-5) (Table 2). The ^13^C NMR data of **3** accompanied with its phase-sensitive HSQC spectrum exhibited a methyl carbon at δ_C_ 21.6 (C-6); three methoxyl carbons at δ_C_ 48.4 (C-8), 50.7 (C-7), and 55.4 (C-9); a methylene carbon at δ_C_ 48.4 (C-2); a nonprotonated ketal carbon at δ_C_ 99.4 (C-3); a dioxygenated methine carbon at δ_C_ 102.1 (C-5); and a ketone carbonyl at δ_C_ 202.9 (C-4) (Table 1). Correlations from δ_H_ 1.31 (H-6)/δ_H_ 4.11 (H_-1_) and δ_H_ 4.11 (H-1)/δ_H_ 2.36 and 2.65 (H_2_-2) in the COSY spectrum of **3** together with key correlations from δ_H_ 2.36 and 2.65 (H_2_-2)/δ_C_ 202.9 (C-4), δ_H_ 2.36 (H_a_-2)/δ_C_ 99.4 (C-3), δ_H_ 4.94 (H-5)/δ_C_ 67.4 (C-1) and 202.9 (C-4), δ_H_ 3.18 (H_3_-7)/δ_C_ 99.4 (C-3), δ_H_ 3.31 (H_3_-8)/δ_C_ 99.4 (C-3), and δ_H_ 3.39 (H_3_-9)/δ_C_ 102.1 (C-5) in the HMBC spectrum of **3** (Figure 2) established the planar structure of **3**. The relative configurations of C-1 and C-5 in **3** were deduced to be *R** and *S**, respectively, based on a key correlation of δ_H_ 4.11 (H-1)/δ_H_ 3.39 (H_3_-9) in the NOESY spectrum of **3** (Figure 2). H_3_-8 and H-5 were determined to be located at the same side due to a cross-peak of δ_H_ 3.31 (H_3_-8)/δ_H_ 4.94 (H-5) in the NOESY spectrum (Figure 2). Since compound **3** was speculated reasonably to be derived originally from (*R*)-3-hydroxybutyric acid (**6**) and glycolic acid and was further synthesized via sequential enolyzation, condensation, acyloin rearrangement [19], and methylation (Figure 1), the absolute configurations of its C-1 were thus assigned as the *R* form and the C-5 was then established as the *S* form.

The ^1^H, ^13^C, and HSQC data of compound **4** were almost identical to those of compound **3** except that a methylene at C-2 and a methoxyl group at C-3 in **3** was replaced by an olefinic functionality at [δ_H_ 6.02 (d, *J* = 1.8 Hz, H-2); δ_C_ 121.2 (C-2)] and δ_C_ 148.2 (C-3) in **4**. Complete assignments of COSY, HMBC, and NOESY spectra of **4** (Figure 2) allowed the elucidation of its planar structure as shown in Figure 1. That was further corroborated by a quasi-molecular ion [M + H]^+^ at *m/z* 173.0808 (calcd. 173.0814 for C_8_H_13_O_4_) in the HRESIMS and a carbonyl signal at 1739 cm^−1^ in the IR spectrum of **3** shifted to 1707 cm^−1^ in that of **4** due to olefinic conjugation effect. Compound **4** was also inferred to originate from compound **6** and glycolic-acid-like compound **3** (Figure 1), and the absolute configurations of C-1 and C-5 in **4** were deduced to be the same as those of **3**.

The molecular formula of compound **5** was deduced to be C_9_H_14_O_6_ by a quasi-molecular ion [M − H]^−^ at *m*/*z* 217.0715 (calcd. 217.0711 for C_9_H_13_O_6_) and supported by its ^13^C NMR data (Table 1), indicating a double bond equivalence (DBE) value of three. The ^1^H and ^13^C NMR data of A ring (C-1–C-6) of **5** were consistent with those of **3** except that the carbonyl (δ_C_ 202.9, C-4) in **3** disappeared (Figure 2), and instead a hemiketal carbon signal δ_C_ 93.1 (C-4) in **5** was observed in addition to three methoxyl groups (δ_H_ 3.18, 3.31, and 3.39; δ_C_ 48.4, 50.7, and 55.4) in **3** replaced by two oxygenated methylenes (δ_H_ 3.72, 4.03, 4.07, and 4.34; δ_C_ 63.6 and 69.2) and an oxygenated methine (δ_H_ 3.77; δ_C_ 72.5) in **5** (Table 1 and Table 2). Key cross-peaks from δ_H_ 1.14 (H_3_-6)/δ_H_ 4.38 (H-1) and δ_H_ 4.38 (H-1)/δ_H_ 1.67 and 1.79 (H_2_-2) in the COSY spectrum along with key correlations of δ_H_ 1.67 and 1.79 (H_2_-2)/δ_C_ 89.4 (C-3) and 93.1 (C-4) and δ_H_ 4.62 (H-5)/δ_C_ 66.6 (C-1) and 93.1 (C-4) established the ring A of **5** as shown in Figure 2. The other signals at [δ_H_ 3.72 (d, *J* = 12.0 Hz, H_a_-7), 3.77 (t, *J* = 3.0 Hz, H-8), 4.03 (d, *J* = 12.0 Hz, H_a_-9), 4.07 (dt, *J* = 12.0, 3.0 Hz, H_b_-9), and 4.34 (dt, *J* = 12.0, 3.0 Hz, H_b_-7)] as well as [δ_C_ 63.6 (C-7), 72.5 (C-8) and 69.2 (C-9)] observed in the ^1^H and ^13^C NMR spectra of **5**, respectively, were attributed to be a set of glycerol moieties. Long-range correlations from δ_H_ 4.03 (H_a_-9) and 4.07 (H_b_-9)/δ_C_ 98.8 (C-5) and δ_H_ 3.77 (H-8)/δ_C_ 93.1 (C-4) in the HMBC spectrum confirmed the existence of ring B of **5** (Figure 2), and C-7 was thus proposed to be connected with C-3 via an ether linkage to form ring C to fit the DBE value of **5**. For determining the absolute configuration of compound **5** in this study, a single-crystal X-ray diffraction experiment with Cu *Kα* radiation (λ = 0.154 nm) was employed (Figure 3). The chiralities of C-1, -3, -4, -5, and -8 in **5** were determined to be 1*R*, 3*R*, 4*S*, 5*S*, and 8*R*, respectively, which were consistent with those proposed in the biosynthetic pathway of compound **5** as shown in Figure 1.

### 2.2. Anti-Angiogenesis Activities in Human Endothelial Progenitor Cells

Compounds **1**–**5** were evaluated for anti-angiogenic activity in human endothelial progenitor cells (EPCs) with sorafenib as the positive control [6]. As shown in Table 3, penisterine D (**4**) exhibited inhibition of EPC growth with IC_50_ values of 28.5 ± 2.2 µg/mL. Data from the tube formation and migration assay validated the anti-angiogenic effects of **4** on EPCs. It was found that **4** suppressed the capillary-like tube formation and migration of EPCs (Figure 4A,B, and S40). To determine whether these finding were caused by the potential cytotoxicity of **4**, we measured LDH release by EPCs after **4** treatments. No statistical difference was observed between the control group and EPCs-treated with **4**, which therefore excluded the possibility of cytotoxicity in the anti-angiogenic effect of **4** (Figure 4C). Collectively, these findings reveal that **4** displays the most active anti-angiogenic properties by blocking cell growth, migration, and tube formation of EPCs.

### 2.3. Anti-Angiogenesis Activities in an In Vivo Zebrafish Model

Vascular development in zebrafish is very similar to that of higher vertebrates such as humans, starting during gastrulation and continuing throughout life [20]. Since the amino acid sequences of some genes in humans and zebrafish are highly conserved in vertebrate evolution, the mechanism of human angiogenesis can be explored by studying zebrafish [21]. To monitor the in vivo anti-angiogenesis activity, we applied a transgenic zebrafish *Tg (fli1:EGFP)*, which was expressed EGFP in the vasculature during development. Zebrafish embryos were incubated with **3** and **4** at 1 day post-fertilization (dpf) and evaluated the effect on angiogenesis at 4 dpf. Anti-angiogenesis was grouped into normal, mild, and severe (Figure 5A) according to the effects on intersegmental vessel (ISV) and dorsal longitudinal anastomtic vessel (DLAV) formation. Results showed that the angiogenesis percentages of zebrafish embryos were 54% and 37% by the treatment with 10.2 and 20.4 μg/mL of **3** (Figure 5B), respectively, and 52% and 41% by the treatment with 8.6 and 17.2 μg/mL of **4** (Figure 5D), respectively. We observed that no lethality occurred after 72 h of incubation with **3** and **4** (Figure 5C,E). These data suggest that **3** and **4** showed anti-angiogenic activity.

## 3. Materials and Methods

### 3.1. General Experimental Procedures

Optical rotation, ultraviolet, and IR spectra were measured on a JASCO P-2000 polarimeter (Tokyo, Japan), a Thermo UV–visible Heλios α spectrophotometer (Bellefonte, CA, USA), and a JASCO FT/IR 4100 spectrometer (Tokyo, Japan), respectively. ^1^H and ^13^C NMR spectra were obtained using an Agilent 600 MHz DD2 NMR spectrometer (Agilent Technologies, Santa Clara, CA, USA). High-resolution electrospray ionization mass spectra were obtained using an Orbitrap QE Plus mass spectrometer MS000100 (Thermo Fisher Scientific Inc., Waltham, MA, USA). Sephadex LH-20 (Sigma-Aldrich, St. Louis, MO, USA) was used for open column chromatography. Thin-layer chromatography was performed using silica gel 60 F_254_ plates (0.2 mm) (Merck, Darmstadt, Germany). An L-7100 HPLC pump (Hitachi, Tokyo, Japan) equipped with a refractive index detector (Bischoff, Leonberg, Germany) was employed for compound purification. All spectroscopic data are presented in the Appendix A.

### 3.2. Algal Material

The algal material was collected in July 2021 off the coast of Badouzi (25°08′50.9″ N 121°47′42.3″ E), Keelung, Taiwan. Alga specimen was identified as *Sargassum cristaefolium* by T.-H.L. A voucher specimen (No. SC-IFS-2021) was deposited at Institute of Fisheries Science, National Taiwan University, Taipei, Taiwan.

### 3.3. Isolation and Identification of Fungal Stain

The alga material was soaked in 75% EtOH followed by 0.01% NaOCl_aq_ and treated with ddH_2_O for surface cleaning. The disinfected alga was cut into circles of approximately 5 mm^2^. The sample was placed into the seawater PDA (potato dextrose agar) medium and incubated at 28 °C. A single fungal strain was obtained after continuous separation and purification. The mycelium of fungus was lyophilized and ground. The DNA of powdered material was extracted using DNeasy Plant Mini Kit (Qiagen, Venlo, The Netherlands) following the manufacturer’s protocol. Two sets of primers ITS4 (forward: 5«-TCCTCCGCTTATTGATATGC-3«) and ITS5 (reverse: 5«-GGAAGTAAAAGTCAAGG-3«) were used to amplify the ITS rRNA. The PCR products were analyzed by Genomic Co., Ltd. (New Taipei City, Taiwan). According to BLAST and phylogenetic analysis based on ITS rRNA gene sequences, the strain SC29 was identified as *Penicillium sumatraense*. The sequence was deposited in GenBank under the accession number ON685565. This stain is currently preserved in Institute of Fisheries Science, National Taiwan University, Taipei, Taiwan. We performed a most parsimonious tree (MPT) using 8 species (Figure 6). The evolutionary history was inferred using the maximum parsimony method. Tree #1 out of the 5 most parsimonious trees (length = 77) is shown. The consistency index is 0.961039 (0.938776), the retention index is 0.976923 (0.976923), and the composite index is 0.938861 (0.917111) for all sites and parsimony-informative sites (in parentheses). The percentage of replicate trees in which the associated taxa clustered together in the bootstrap test (1000 replicates) is shown next to the branches [22]. The MP tree was obtained using the tree-bisection-regrafting (TBR) algorithm (p. 126, [23]) with search level 1, in which the initial trees were obtained by the random addition of sequences (5 replicates). This analysis involved 18 nucleotide sequences. There were a total of 629 positions in the final dataset. Evolutionary analyses were conducted in MEGA11 [24].

### 3.4. Extraction and Isolation of Secondary Metabolites

The liquid-state mass cultures of *P. sumatraense* SC29 were carried out in seawater PDB (potato dextrose broth) medium. A single colony of the strain from the agar plate was inoculated into the 250 mL flask containing 100 mL PDB medium and incubated at 28 °C for 14 days on a rotary shaker at 180 rpm. In total, 7.2 L fermentation broth was harvested and partitioned using EtOAc and water three times. The EtOAc extract (3.6 g) was further subjected to size exclusion chromatography on a Sephadex LH-20 column (2.8 cm i.d. × 68 cm) and eluted with 100% EtOH at a flow rate of 2.0 mL/min to give 25 fractions. Each fraction (25 mL) collected was checked for its composition by TLC using DCM/MeOH (10:1) for development, and dipping in vanillin-H_2_SO_4_ was used in the detection of compounds with similar skeletons. All fractions were combined into four portions I–IV. Portion II (frs. 14–16) was rechromatographed on a semipreparative reversed-phase HPLC (Phenomenex Luna 5 μ PFP, 10 × 250 mm) with MeCN/H_2_O (3:7, *v*/*v*) as eluent to yield eight subfractions (F2A–F2H). F2A (1.05 g) was separated on semipreparative HPLC (Phenomenex Luna 5 μ C18, 10 × 250 mm) with 5% MeCN_aq_ containing 0.1% formic acid as eluent to afford **1** (7.12 mg, t_R_ = 14.0 min), **2** (33.62 mg, t_R_ = 27.8 min), **5** (22.7 mg, t_R_ = 19.6 min), and **6** (42.24 mg, t_R_ = 8.9 min). Compound **3** (6.61 mg, t_R_ = 10.0 min) was isolated from F2B by semipreparative reversed-phase HPLC (Phenomenex Luna 5 μ C18, 10 × 250 mm) using 40% MeCN_aq_ as eluent. Compound **4** (12.1 mg, t_R_ = 22.0 min) was isolated from F2D by semipreparative reversed-phase HPLC (Phenomenex Luna 5 μ C18, 10 × 250 mm) using 30% MeCN_aq_ containing 0.1% formic acid as eluent.

*Penisterine A* (**1**): colorless oil; [α]^26^_D_ -9.8 (c = 0.02, MeOH); IR (ZnSe) ν_max_: 3426, 1728, 1387, 1288, 1207, 1138, 1090, 1055 cm^−1^; ^1^H and ^13^C NMR spectroscopic data: see Table 1 and Table 2; HRESIMS *m*/*z* 161.0450 (calcd. 161.0450 for C_6_H_9_O_5_)

*Penisterine B* (**2**): colorless oil; [α^]26^_D_ -14.4 (c = 0.02, MeOH); IR (ZnSe) ν_max_: 3445, 1737, 1439, 1384, 1308, 1266, 1201, 1138, 1096, 1055, 1001 cm^−1^; ^1^H and ^13^C NMR spectroscopic data: see Table 1 and Table 2; HRESIMS *m*/*z* 175.0506 (calcd. 175.0507 for C_7_H_11_O_5_)

*Penisterine C* (**3**): brown oil; [α]^26^_D_ -33.2 (c = 0.02, MeOH); IR (ZnSe) ν_max_: 1739, 1115, 1098, 1062 cm^−1^; ^1^H and ^13^C NMR spectroscopic data: see Table 1 and Table 2; HRESIMS *m*/*z* 205.1069 (calcd. 205.1076 for C_9_H_17_O_5_)

*Penisterine D* (**4**): Yellowish oil; [α]^26^_D_ 59.0 (c = 0.02, MeOH); UV (MeOH) λ_max_ (log ε) 264 (2.7) nm; IR (ZnSe) ν_max_: 2933, 1707, 1632, 1455, 1375, 1318, 1244, 1199, 1172, 1153, 1091, 1057, 992, 970, 848, 832 cm^−1^; ^1^H and ^13^C NMR spectroscopic data: see Table 1 and Table 2; HRESIMS *m*/*z* 173.0808 (calcd. 173.0814 for C_8_H_13_O_4_)

*Penisterine E* (**5**): amorphous white powder; [α]^26^_D_ -40.8 (c = 0.02, MeOH); IR (ZnSe) ν_max_: 3411, 2928, 1641, 1122, 1078, 1031, 975 cm^−1^; ^1^H and ^13^C NMR spectroscopic data: see Table 1 and Table 2; HRESIMS *m*/*z* 217.0715 (calcd. 217.0711 for C_9_H_13_O_6_)

### 3.5. X-ray Diffraction Analysis

The crystal data were acquired on an Oxford Gemini Dual System diffractometer. The data of compound **5** were acquired with Cu *K*α radiation, and the crystal data and experimental details are listed in Appendix A.

Crystallographic Data for Compound **5**. (CCDC 2180439) The crystal was obtained from methanol-n-hexane–acetone (4:2:1). Crystal data: a = 5.9346(2) Å, b = 10.4559(4) Å, c = 15.4630(6) Å, α = 90°, β = 90°, γ = 90°, V = 959.50(6) Å3, μ(Cu *K*α) = 1.102 mm^−1^. Flack parameter = 0.12(5).

### 3.6. Alkaline Hydrolysis of **1**

Compound **1** (10.0 mg) and 2 mL of a CH_3_OH/0.5 M NaOH (1:1) were mixed and stirred at room temperature for 15 h, then the reaction mixtures were dried using rotary evaporator to remove CH_3_OH and neutralized by the addition of HCl. The reaction mixtures were then extracted with EtOAc, and the organic layer was evaporated in vacuo, and the residue was purified by HPLC (Phenomenex Luna 5 μ PFP, 10 × 250 mm) using MeCN/H_2_O containing 0.1% formic acid (5:95) with flow rate of 2 mL/min as eluent to afford (R)-3-hydroxybutanoic acid (**6**) (3.7 mg, t_R_ = 14.8 min).

*(R)-3-Hydroxybutanoic acid (**6**)*: clear oil; [α]^26^_D_ -20.2 (c = 0.02, MeOH); ^1^H NMR (600 MHz, CD_3_OD): δ_H_ 1.20 (3H, d, *J* = 6.0 Hz), 2.40 (2H, m), 4.15 (1H, m); ^13^C NMR (150 MHz, CD_3_OD): δ_C_ 23.4, 44.7, 65.7, 175.6.

### 3.7. Isolation and Cultivation of Human EPCs

Human EPCs were isolated and cultured by the protocols as previously described [25]. Ethical approval for the collection of human EPCs was granted by the Institutional Review Board of Mackay Medical College, New Taipei City, Taiwan (P1000002).

### 3.8. Cell Growth Assay

EPCs were cultured in 96-well plates at a density of 5 × 10^3^ cells in each well. After 24 h of incubation, the culture medium was replaced with fresh MV2 complete medium containing 2% FBS in the presence of either vehicle (DMSO) or compounds. After 48 h of treatment, the survival rate of EPCs was assayed by SRB staining according to previously described procedure [6].

### 3.9. Capillary Tube Formation Assay

The capillary tube formation assay was carried out on Matrigel-coated 96-well plates. EPCs were seeded at the density of 1.25 × 10^4^ cells per well and incubated in MV2 complete medium with 2% FBS and the indicated concentration of tested compound for 24 h at 37 °C. EPCs differentiation and capillary-like tube formation was performed in three wells for each condition. The long axis of each tube was measured with MacBiophotonics Image J software in 3 randomly chosen fields per well.

### 3.10. Cell Migration Assay

Transwell inserts (8 μm pore size, Costar, NY, USA) were used for migration determination. EPCs migratory ability was assayed by the method based on our previous work [26].

### 3.11. Cytotoxicity Assay

EPCs (5 × 10^3^ cells/well) were seeded onto 96-well plates and incubated with MV2 complete medium containing 2% FBS in the presence of vehicle (DMSO) or penisterine D. Release of lactate dehydrogenase (LDH) into the medium was measured using a cytotoxicity assay kit (Promega, Madison, WI, USA).

### 3.12. Zebrafish

Zebrafish (Danio rerio) and embryos were maintained at 28 °C. All animal procedures were approved by the Institutional Animal Care and Use Committee or Panel (IACUC/IACUP) (protocol No.: LAC-2021-0181). The methods were carried out in accordance with the approved guidelines.

### 3.13. Transgenic Zebrafish Lines

The transgenic zebrafish line *Tg (fli1:EGFP)* was used in this study. The *Tg (fli1:EGFP)* containing fli1 (friend leukemia integration 1 transcription factor, 15 kb) promoter, driving the expression of enhanced green fluorescent protein (EGFP) in all blood vessels throughout embryogenesis [27], enables anti-angiogenesis readout for drug treatment.

### 3.14. Embryo Collection

One day prior to fertilization, male and female adult zebrafish were placed individually into mating tanks with inner mesh. Male and female fish were separated by a separator and left in mating cages overnight. The next morning after the removal of the separator, the couple zebrafish stimulated by the light started to chase each other and lay eggs and sperm. After 1 h, the embryos were collected and transferred to a 100 mm dish with E3 solution (5 mM NaCl, 0.17 mM KCl, 0.33 mM CaCl_2_, and 0.33 mM MgSO_4_, pH 7.0) [28] and incubated at 28 °C for 6 h. The unfertilized and dead embryos were removed, and the remaining live embryos were replenished with fresh E3 solution and kept for incubation.

### 3.15. Angiogenesis Inhibition Drug Screening Platform

At 1 day post-fertilization (dpf), the *Tg (fli1:EGFP)* embryos were distributed into 6-well culture plates with 25 embryos per well containing 3 mL E3/PTU (1-phenyl-2-thiourea, 0.003%) buffer. The embryos were treated with drugs at various concentrations. The embryos were anesthetized with tricaine (ethyl 3-aminobenzoate methanesulfonate, MS-222 (Sigma-Aldrich In., St. Louis, MO, USA) final concentration 0.016%) to prevent movement. Antiangiogenic effects of embryos were analyzed and images were captured with a fluorescence phase-contrast Zeiss Axio Vert.A1 inverted microscope (Zeiss, Jena, Germany) and a Leadview 2800AM-FL camera (Leadview, Taipei, Taiwan) at 4 dpf. Sorafenib was used as a positive control and DMSO (0.1%) was used as a negative control.

### 3.16. Survival Test

*Tg (fli1:EGFP)* embryos were used in the survival assay. At 4 dpf, 25 embryos were placed into 1 well of the 6-well plates wih 3 mL E3 medium supplement with drugs. The DMSO control and different compounds were serially diluted to determine the survival rate. Two days after exposure, the embryos were counted and the survival curves were measured. The heartbeat was used to evaluate mortality of zebrafish.

### 3.17. Statistical Analysis

Statistical analysis was performed from three independent experiments and analyzed the mean ± standard deviation (SD). One-way analysis of variation (AVOVA) followed by Tukey’s test was used to analyze the statistical significance and indicated by * *p* < 0.05, ** *p* < 0.01, and *** *p* < 0.001.

## 4. Conclusions

As a result, previously unreported compounds penisterine A (**1**), penisterines C–E (**3**–**5**), and penisterine A methyl ether (**2**) were isolated for the first time from natural resources with a known compound and were identified from a marine alga-derived fungus *Penicillium sumatraense* SC29. Among these, penisterine E (**5**) was a unique 6/6/6-tricyclic ether containing an acetal and two hemiketal functionalities. In addition, a possible biosynthetic pathway of **1**–**5** from the known compounds, (*R*)-3-hydroxybutanoic acid (**6**) and glycolic acid, was proposed. Penisterine D (**4**) shows anti-angiogenesis activity in both human EPCs and a Tg zebrafish model. The angiogenesis activity of penisterine D (**4**) makes it an attractive candidate for further preclinical investigation. Although penisterine C (**3**) did not have a significant effect on EPC at the dose seen with penisterine D (**4**), it did possess an anti-angiogenic effect in zebrafish. Thus, further investigation is required to understand the mechanism for the ability of penisterine C (**3**) to inhibit vessel development in embryonic zebrafish.

## Data Availability

Not applicable.

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
