# Peer review of "Chemical Constituents and Anti-Angiogenic Principles from a Marine Algicolous Penicillium sumatraense SC29"

_molecules, 2022, doi:10.3390/molecules27248940_

Round 1
Reviewer 1 Report
In the manuscript by Shi et al. entitled “ Chemical Constituents and Anti-angiogenesis Principles from a Marine Algicolous Penicillium steckii SC29” the authors isolate a new compound that is shown to inhibit angiogenesis. The authors isolated Penicillium steckii SC29 from marine brown alga. From the P. Steckii SC29 the authors extracted and identified five (R)-3-hydroxybutyric and glycolic acid derivatives. The compounds were characterized using 1N and 13C NMR, Mass Spec and x-ray diffraction. The 5 isolated compounds were analyzed for anti-angiogenic activity on human EPC isolated from blood using a cell growth assay. Compound 4 (Penisterine D) was the only compound to inhibit EPC growth Thus, the authors focused the remaining anti-angiogenic experiments on compound 4. Using the Matrigel tube formation assay compound 4 inhibited the formation of tubes in a concentration dependent manner. Cytotoxicity of compound was analyzed using LDH release. Lastly, Zebrafish embryos were treated with compound 4 and shown to have inhibitory effects on the development of intersegmental vessels. The authors provide initial evidence that Penisterine D is an anti-angiogenic agent.
Major Issues:
1. The authors should test the other 4 agents for anti-angiogenic activity. The uses of a proliferation assay (fig 1) to rule out anti-angiogenic activity is flawed. This assay only indicated that compound 4 inhibits mitotic activity and compounds 1-3 and 5 do not. There are several pathways (migration, VEGFR signaling, etc.) that can be inhibited to prevent angiogenesis. Thus, compounds 1-3 and 5 could have anti-angiogenic properties even though they do not inhibit mitotic activity.
2. The authors should perform a migration assay (scratch or transwell) for all compounds to see if inhibition of motility is a possible mechanism of action for anti-angiogenic activity.
3. The authors use LDH release as a test for cell death. This assay is useful in determining membrane disruption and is a test that better measures of necrosis than apoptosis. Also, the authors do not indicated length of incubation only concentration used for the LDH assay. This information should be added to the Results section. The analysis of Annexin V expression should be performed as it is a better test for cell death (apoptosis and necrosis) and its expression is an early cell death marker.
4. The authors should test all the compounds on vessel development of the Zebrafish embryo. Compounds 1-3 and 5 may affect angiogenesis through a different pathway than compound 4.
5. A cell death assay should be performed on the Zebrafish embryos treated with the compounds to determine if anti-angiogenic activity is due to cell death or inhibition of a different mechanism.
Minor Issues:
1. Provide the primer sequences for ITS4 and ITS5 in Materials and Methods
2. Provide the number of wells imaged for each condition in the tube formation assay. Only provide the number of fields/well (Materials and Methods).
3. Indicate how survival was determined for the survival assay (Materials and Methods)
Reviewer 2 Report
Comments:
1. The manuscript depicts a fungal identification was done − Penicillium steckii SC29. However, the author didn’t provide a strong explanation that the fungus was clearly identified.
- Please provide the colony and mycelium characteristics of the isolated fungus. The pictures of the isolated fungus both macroscopic and microscopic are required. Please provide.
- Especially, the PCR bands and phylogenetic tree of the identified Penicillium steckii SC29 are important. Please provide in the revised manuscript.
2. Line 245. Please add the reference of positive control.
3. Section 3.4, line 327. Please check again the information: compound 6 was isolated from fraction F2A or was a product of alkaline hydrolysis of compound 1?
Also, there are some issues need to revise:
1. Line 46. “anti-gngiogenesis”--- “anti-angiogenesis”
2. Line 111. “3400-2400 cm -1”
3. Line 112. “a methyl protons” --- “a methyl proton”
4. Line 155. the presence of “an” additional methyl group …
5. Line 172. “a methylene carbon”
6. Line 194. “a methylene at C-2”
Round 2
Reviewer 1 Report
The authors performed all the suggest experiments to determine whether compounds 1-3 and 5 had anti-angiogenic activity. The authors demonstrated that compounds 1, 2 and 5 did not show anti-angiogenic activity. The authors do show that compound 3 inhibits vessel growth in zebarfish similar to that of compound 4 (supl. 43) this data should be included in figure 5. Even though compound 3 does not inhibit EPC function (migration or tube formation) the compound does inhibit vessel formation in zebrafish. Although compound 3 may not have the same inhibitory affects as compound 4 the authors demonstrate it does have an anti-angiogenic effect, this should be shown in figure 5 not as supplemental material.
The authors should indicate in the conclusion that although compound 3 did not have a significant effect on EPC at the dose seen with compound 4 it did possess an anti-angiogenic effect in zebrafish. Thus, further investigation is required to under the mechanism for the ability of compound 3 to inhibit vessel development in embryonic zebrafish.
Author Response
Thanks for your comment. This information has been added in Materials and Methods (Figures 5B and 5C) and Conclusion section.